# Prevalence and factors associated with peripheral neuropathy in a setting of retail pharmacies in Malaysia–A cross-sectional study

Siew Mooi Ching[1,2]*, Kai Wei Lee[3], Abdul Hanif Khan Yusof Khan[4], Navin Kumar Devaraj[1], Ai Theng Cheong[1], Sook Fan Yap[5], Fan Kee Hoo[6], Wan Aliaa Wan Sulaiman[6], Wei Chao Loh[6], Shen Horng Chong[1], Mansi Patil[7], Vasudevan Ramachandran[8,9,10]

1 Department of Family Medicine, Faculty of Medicine and Health Sciences, Universiti Putra Malaysia, Serdang, Selangor, Malaysia, 2 Department of Medical Sciences, School of Healthcare and Medical Sciences, Sunway University, Bandar Sunway, Malaysia, 3 Department of Medical Microbiology, Faculty of Medicine and Health Sciences, Universiti Putra Malaysia, Serdang, Selangor, Malaysia, 4 Department of Neurology, Faculty of Medicine and Health Sciences, Universiti Putra Malaysia, Serdang, Selangor, Malaysia, 5 Department of Pre-Clinical Sciences, Faculty of Medicine and Health Sciences, Universiti Tunku Abdul Rahman, Kajang, Selangor, Malaysia, 6 Department of Neurology, Faculty of Medicine and Health Sciences, Universiti Putra Malaysia, Serdang, Selangor, Malaysia, 7 Department of Nutrition, Asha Kiran JHC Hospital, Pune, Maharashtra, India, 8 Department of Medical Sciences, Faculty of Health Sciences, University College of MAIWP International, Taman Batu Muda, Batu Caves, Kuala Lumpur, Malaysia, 9 Department of Conservative Dentistry and Endodontics, Saveetha Dental College and Hospitals, Saveetha Institute of Medical and Technical Sciences, Saveetha University, Chennai, India, 10 Malaysian Research Institute on Ageing, Faculty of Medicine and Health Sciences, Universiti Putra Malaysia, Serdang, Selangor, Malaysia

* sm_ching@upm.edu.my

**Data Availability Statement:** All relevant data are uploaded under UPM repository website and the license applied to the uploaded data is CC BY 4.0

## Abstract

Peripheral neuropathy is a common cause for neurological consultation, especially among those with diabetes mellitus. However, research on peripheral neuropathy among the general population is lacking in Malaysia. This study aimed to determine the prevalence and factors associated with peripheral neuropathy in a setting of retail pharmacies. This cross-sectional study of 1283 participants was conducted at retail pharmacies in Selangor. Peripheral neuropathy was defined as the final score in the mild to severe category in the severity rating scale using a biothesiometer. SPSS version 26 was used to perform the analysis. Multiple logistic regressions were used to determine the factors associated with peripheral neuropathy. The prevalence of peripheral neuropathy based on the biothesiometer was 26.5%. According to multiple logistic regression, the predictors of peripheral neuropathy were those who have diabetes (AOR = 3.901), aged more than 50 years (AOR = 3.376), have secondary education or below (AOR = 2.330), are male (AOR = 1.816), and have underlying hypertension (AOR = 1.662). Peripheral neuropathy is a reasonably prevalent condition, affecting a quarter of the general population, and often goes undiagnosed. It is crucial for healthcare providers to proactively screen for peripheral neuropathy, particularly in high-risk populations, to prevent potential complications.

compliant. (http://putrarepo.upm.edu.my/direct/UPM-NL-216).

**Funding:** This research was funded by Procter & Gamble (M) Sdn. Bhd (Vote ID: 6380068). The funder had no role in study design, data collection and analysis, the decision to publish or the preparation of the manuscript.

**Competing interests:** Authors are required to disclose financial or non-financial interests that are directly or indirectly related to the work submitted for publication.

## Introduction

Peripheral neuropathy presents with a range of symptoms which are often vague [1]. The presence of peripheral neuropathy could result in a tremendous impact on daily movement especially walking, running, climbing staircases and sleeping [2]. Peripheral neuropathy can result from various reasons, one of the most common of which is diabetes, as nerves can become damaged by chronically high blood sugar leading to inflammation, oxidative stress, mitochondrial dysfunction and cell death [3]. However, the disease is typically underdiagnosed, as some patients are asymptomatic [4].

According to International Diabetes Federation (IDF) Diabetes Atlas 2021, 537 million adults (20–79 years) are living with diabetes worldwide (1 in 10). South-East-Asia and Western Pacific regions account for half of all diabetes cases (296 million). The total number of people with diabetes is predicted to rise to 643 million (1 in 9 adults) by 2030 and 784 million (1 in 8 adults) by 2045 globally (IDF, 2022). In Malaysia, the prevalence of diabetes among adults in Malaysia was 14.9% in 2006, which has increased to 17.5% in 2015 [5, 6] and further raised to 18.3% in 2019 [7] and 19% in 2021 [8].

Studies have shown that type 2 diabetes mellitus patients have a 15–25% lifetime risk of developing foot ulcers. Early detection of diabetic peripheral neuropathy (DPN) is imperative to reduce the associated morbidity, which can have negative effects on quality of life, work productivity and healthcare resources [9–11]. A study reported that 35% of patients with type 2 diabetes had peripheral neuropathy [12] and were diagnosed with Neuropathy Symptom Score and Neuropathy Deficit Score. A study among patients with type 2 diabetes mellitus attending a follow-up visit in an outpatient clinic at University Kebangsaan Malaysia Medical Center, Kuala Lumpur, Malaysia found that the prevalence of DPN was 79.1% [13] (based on the Neuropathy Disability Score). Another study in the Primary Care Clinic, Universiti Hospital, Kuala Lumpur, Malaysia which included 138 diabetic patients were assessed using the neuropathy symptom score and neuropathy disability score, reported that the prevalence of DPN was high at 50.7% [14], and another study reported 54.1% of DPN based on nerve conduction study [12]. Overall, the findings from a recent meta-analysis showed that the pooled prevalence of DPN was 30% and DPN is more prevalent in people with T2DM (31.5%) compared to those in T1DM (17.5%) [15]. Prevalence of peripheral neuropathy in patients with diabetes: A systematic review and meta-analysis. Primary care diabetes, 14(5), 435–444. Another meta-analysis also reports that the DPN was also common (>10%) among those with pre-diabetes [16]. However, the prevalence of painful diabetic neuropathy was relatively low (5.4%) (based on Neuropathy Symptom Score and Neuropathy Disability Score) in one of the studies conducted locally in a university hospital [17].

In Malaysia, DPN is one of the most common complications of diabetes mellitus, and may lead to foot gangrene, amputation, and neuropathic pain [18]. Lack of widespread awareness about peripheral neuropathy, coupled with its subtle symptoms that do not significantly disrupt daily life, often results in individuals with this condition neglecting to seek medical attention for these minor manifestations. The prevalence of peripheral neuropathy among the general population worldwide was 2.4% [19]. Diabetes mellitus is a highly prevalent condition in the Malaysian population, and diabetic neuropathy is one of the earliest and most common complications arising from it. As a large segment of the population is affected by diabetes, it is reasonable to assume that diabetic neuropathy is likely the most common form of neuropathy encountered in the general population. In contrast, other types of neuropathies mentioned, such as Guillain-Barré syndrome (GBS), chronic inflammatory demyelinating polyneuropathy (CIDP), amyloid neuropathy, and acute inflammatory neuropathies, are relatively rare and often present with rapidly progressive or severe symptoms that would prompt individuals to

seek medical attention promptly. These forms of neuropathy are typically more severe and require immediate medical intervention, making them less relevant for the specific focus of this paper, which primarily addresses early detection and monitoring of neuropathy in the general population [15–17].

To date, a study on peripheral neuropathy in the general adult population is lacking in Malaysia.

The monofilament test, ankle reflex, and vibration perception testing using a 128-Hz tuning fork are the recommended and commonly used tools for screening and diagnosing peripheral neuropathy. However, it is important to note that the monofilament test is primarily effective in detecting moderate-to-severe PN. In contrast, vibration perception threshold testing measures the integrity of large nerve fibers by converting an electric current into a transverse vibration mode for the patient. Perception is typically poorer in the lower extremities than in the upper extremities, making vibration perception threshold a crucial factor in the early detection of diabetic peripheral neuropathy and consequently reducing the risk of foot ulceration [2, 20, 21]. The objective of this study was to establish the prevalence of peripheral neuropathy among the general population in Malaysia and identify the factors associated with it using a biothesiometer. By doing so, this study seeks to raise public awareness regarding peripheral neuropathy and emphasize the significance of early detection through biothesiometry.

## Methods

### Study design, study population and sampling method

This cross-sectional study was conducted between March 15, 2021 and May 5, 2022 at 7 retail pharmacies in Malaysia. Retail pharmacies were selected as data collection sites based on several considerations. Firstly, individuals with chronic diseases, such as diabetes, frequently visit retail pharmacies to obtain their medications or seek alternative therapeutic options like dietary supplements. These individuals are already engaged with pharmacists in managing their health conditions. By conducting this study at retail pharmacies, we could access a significant portion of the target population who are likely to be more health-literate and inclined to learn about their health status.

Additionally, we conducted this study in the form of a health campaign offering free neuropathy screenings, which further facilitated convenient data collection. This approach not only benefits those with diabetes by providing them with an opportunity to assess their neuropathy status, but also allows individuals without diabetes to undergo screening and increase their awareness about neuropathy. Furthermore, the health campaign raised awareness among the general public visiting the pharmacy, including family members or companions of those with diabetes.

The inclusion criteria for this study were that the participants must be Malaysian, aged ≥18 years old, and willing to sign the informed consent form. Those critically ill and/or mentally challenged were not eligible to participate in this study. Participants were recruited using a convenient sampling method.

### Sample size calculation

The sample size was calculated using the StatCalc function in Epi Info 7.0, based on the prevalence of peripheral neuropathy among the Parsi community of Bombay, which was 2.38% [22]. The estimated sample size was 899, with a 95% confidence interval (CI) with a power of 80% and a p-value <0.05. The total number of participants needed was 1283, after taking into account a missing value rate of 30%. While determining the appropriate sample size for our study on the prevalence of diabetic neuropathy in the community, we encountered a scarcity

of recent, locally relevant research to serve as a reference. The paucity of community-based neuropathy studies, particularly in our region, necessitated the use of an older study conducted in Mumbai, India, in 1991 [22]. Despite its age, this study remains one of the few comprehensive investigations into the prevalence of neuropathy in a community setting, making it a valuable resource for our sample size calculations.

For the factors associated with peripheral neuropathy, we used G*Power software, which calculated a required sample size of 8,422. This calculation was based on a study conducted at Universiti Kebangsaan Malaysia, in which older age was identified as a determinant of diabetic peripheral neuropathy among patients with diabetes (OR 1.13, 95% CI 1.01–1.26, p = 0.039) [13]. Due to the impractically large sample size and time constraints, we adopted a sample size of 899 for this study.

## Data collection tools

There were four sections in the questionnaire. The first section was used to capture socio-demographic information (i.e. age, gender, ethnicity, and personal monthly income in Malaysian Ringgit); lifestyle (i.e. alcohol consumption, smoking, and vegetarianism); and co-morbidities (i.e. hypertension, diabetes, neurological disorder, and family history of the neurological disorder). If they reported having diabetes, we asked about how long they had been diagnosed with diabetes and whether they experienced any complications due to diabetes, such as stroke, heart disease, renal disease, eye disease or foot ulceration. Other sections covered peripheral neuropathy screening tests, neuropathy symptom scores and neuropathy disability scores.

## Peripheral neuropathy screening test

The peripheral neuropathy test was the determination of the vibration perception threshold on both feet using the digital biothesiometer by P&G (Diabetik Foot Care Model: Vibrometer-VPT model 1; The Digital 0 to 50 Volts indicator with a portable Vibration probe functioning at 230V, +/- 20%, AC, 50Hz Mains operation) [23]. The biothesiometer probe can vibrate with an amplitude proportional to the square of the applied voltage. To test the vibration perception threshold, a vibration probe must be placed on six sites on each foot. The sites are the plantar aspects of the tip of the first toe, the base of the first, third and fifth toes, the medial aspect of the midfoot and at the heel.

After patients were familiarised with the sensation by holding the probe against the distal palmar surface of the hand, the probe was then applied perpendicular to the distal plantar surface of the big toe of both the legs. The voltage slowly increased at the rate of 1 mV/sec, and the vibration perception threshold value is defined as the voltage level when the patient indicates that he or she first feels the vibration sense. The mean of three readings at each site was taken, with a higher vibration unit value indicating poorer performance or greater sensory dysfunction.

## Symptoms and sensory examination

A physical examination was also conducted to determine the neuropathy disability score on the feet, which included testing the ankle reflex, vibration perception, pinprick sensation, and feet temperature by touch. Then, we furthered the workup by asking questions on neuropathy symptoms, such as whether they were experiencing sensations of burning/numbness/tingling or fatigue/cramping/aching. If they had one of those symptoms, patients were asked where the symptoms occurred, whether the feet, calves, or elsewhere on the foot. Furthermore, we also asked if symptoms worsened during the day at night or both, and how they relieve symptoms,

whether by walking, standing, or sitting/lying down. The scoring methods for the neuropathy symptoms scale and neuropathy disability scale are shown in Appendix 1 and 2. In brief, the sum of neuropathy symptom score ranges from 0–9, whereby the severity of peripheral neuropathy can be categorized into normal (sum of score 0–2), mild symptoms (sum of the score of 3–4), moderate symptoms (sum of score 5–6) and severe symptoms (sum of score 7–9). For the neuropathy disability score, the total score ranges from 0–10. The total score can be used to indicate the severity of sign of peripheral neuropathy, whereby normal (sum of score 0–2), mild signs of peripheral neuropathy (sum of score 3–5), moderate signs (sum of score 6–8), and severe signs (sum of score 9–10) [24].

## Statistical analysis

All analyses were conducted using the Statistical Package for the Social Sciences version 23.0. Descriptive statistics were computed, obtaining mean and standard deviation (SD) or median and interquartile range (IQR) for the baseline characteristics of the participants. The association between the independent variables (i.e. age, gender, ethnicity, education level, alcohol consumption, smoking, vegetarian, hypertension, diabetes, neurological disease, and family history of neurological disease). The dependent variable for peripheral neuropathy was classified as "yes" for individuals with mild to severe peripheral neuropathy, while "no" refers to those with normal conditions, underwent testing through either the Chi-Square test or Fisher's exact test. Variables with a p-value $< 0.25$ in the simple logistic regression (derived from the results in the normal versus mild-severe column) were entered into multiple logistic regressions to look for determinants of peripheral neuropathy (Mild-severe). The level of significance is set at a p-value $< 0.05$.

## Operational definition

Peripheral neuropathy is classified as "yes" when there is present of mild to severe symptoms of peripheral neuropathy. Conversely, it is categorized as "no" when it fulfils the normal range of peripheral neuropathy symptoms. The severity of peripheral neuropathy based on biothesiometer test has been categorised as normal (1-15v), mild (16-20v), moderate (21-25v), and severe (26-50v) for those above 50 years old; for those 50 years old and below, normal is 1-10v, mild neuropathy (11-15v), moderate neuropathy (16-20v) and severe neuropathy (21-50v) [25].

## Ethical approval

Ethics approval was obtained from the Medical Research Ethics Committee (MREC), Ministry of Health Malaysia (NMRR-20-971-54860) and Ethics Committee for Research Involving Human Subjects, Universiti Putra Malaysia (JKEUPM-2020-367). The study's approaches that involved human participants were in accordance with the ethical standards set by the institutional and national research committee, as well as the 1964 Helsinki declaration and its subsequent amendments or similar ethical guidelines. Prior to data collection, written informed consent was obtained from the respondents, as well as from the parents/legal guardians of uneducated subjects involved in this study.

## Results

### Participants' characteristics

A total of 1283 participants were recruited into this study. The mean age of the participants was 40.6 ± 12.9 years old. About half were Chinese (54.1%), and 43.4% had tertiary education.

The majority did not drink alcohol (80.6%), 83.5% were non-smokers, and 97.6% were non-vegetarian. The percentage of hypertension (21.8%) was more than the percentage of diabetes (12.9%) among the participants, while 3.2% had underlying neurological problems and 7.3% had a family history of any neurological problems. Table 1 shows the characteristics of the participants involved in this study.

## Severity of peripheral neuropathy

As shown in Table 2, 136 (10.6%) participants were found to have severe peripheral neuropathy screened with biothesiometer, 114 participants (11.2%) with mild and 60 participants (4.7%) with moderate peripheral neuropathy. Among those aged ≤50 (n = 749), the percentage of mild, moderate and severe peripheral neuropathy was 8%, 2% and 2.5%, respectively. The rates of mild, moderate and severe peripheral neuropathy were much higher seen in those aged >50 (n = 534), which were 15.7%, 8.4% and 21.9%, respectively.

## Variables associated with peripheral neuropathy

Table 3 shows the association between the characteristics of participants and severity of peripheral neuropathy using the univariate analysis. Age (p<0.001), gender (p<0.001), ethnicity (p = 0.001), education level (p<0.001), smoking (p = 0.012), either hypertension (p<0.001), diabetes (p<0.01) or the other neurological disease (p = 0.065) were found to be significantly associated with peripheral neuropathy.

## Predictors for peripheral neuropathy using multiple logistic regression analysis

Table 4 shows the predictors of peripheral neuropathy using multiple logistic regression analysis. Those who have diabetes [adjusted odd ratio (AOR) = 3.901, 95% CI = 2.610, 5.830), those were aged >50 years old (AOR = 3.376, 95% CI = 2.447, 4.658)], had secondary education and below (AOR = 2.330, 95% CI = 1.709, 3.178), being a male (AOR = 1.816, 95% CI = 1.354, 2.436). Those with hypertension (AOR = 1.662, 95% CI = 1.174, 02.351) had greater odds of having mild to severe symptoms of peripheral neuropathy.

## Discussion

The study found that the overall prevalence of peripheral neuropathy was 26.5%, with a prevalence of 12.6% among participants aged 50 years old or younger and 46.1% among those above 50 years old. Additionally, the prevalence of mild to severe symptoms of peripheral neuropathy in this study was significantly higher than a multinational study that focused only on participants with diabetes from 38 countries worldwide, which reported an overall prevalence of 7.7%, the highest among nations in the South-East Asia region at 9.6%, followed by European nations at 9.4%, Eastern Mediterranean nations at 8.3%, and less than 6% in Western Pacific, African, and American nations [26]. To compare the study's findings with existing literature, the researchers explored the prevalence of peripheral neuropathy among participants with diabetes, which was found to be 66.9% in this study. Moreover, the study found that 20.5% of participants without diabetes had peripheral neuropathy, higher than a cohort study of adults in the USA conducted by Hicks et al., who reported that 11.5% of adults without diabetes had peripheral neuropathy [27]. The higher prevalence of peripheral neuropathy in our study population compared to global studies could be attributed to the biothesiometer's higher sensitivity in detecting early or mild peripheral neuropathy, as well as the possibility of many

**Table 1. Characteristics of participants' socio-demographics, lifestyle, medical background, family history, and diabetic-related complications (n = 1283).**

| Characteristics | Categories | | Aged ≤50 (n = 749) | Aged >50 (n = 534) | All |
|---|---|---|---|---|---|
| Age, in years | Mean (SD) | | 34.8 (8.4) | 58.5 (6.0) | 40.6 (12.9) |
| | Min to Max | | 18–50 | 51–80 | 18–80 |
| | Median ± IQR | | 34 ± 15 | 57 ± 9 | 39 ± 21 |
| Gender, n (%) | Male | | 396 (52.9) | 252 (47.2) | 648 (50.5) |
| | Female | | 353 (47.1) | 282 (52.8) | 635 (49.5) |
| Ethnicity, n (%) | Malay | | 284 (37.9) | 88 (16.5) | 372 (29.0) |
| | Chinese | | 320 (42.7) | 374 (70.0) | 694 (54.1) |
| | Indian | | 101 (13.5) | 60 (11.2) | 161 (12.5) |
| | Others | | 44 (5.9) | 12 (2.2) | 56 (4.4) |
| Highest Education level achieved, n (%) | None | | 12 (1.6) | 27 (5.1) | 39 (3.0) |
| | Primary school | | 21 (2.8) | 105 (19.7) | 126 (9.8) |
| | Secondary school | | 199 (26.6) | 247 (46.3) | 446 (34.8) |
| | Pre-University | | 84 (11.2) | 31 (5.8) | 115 (9.0) |
| | Tertiary level | | 433 (57.8) | 124 (23.2) | 557 (43.4) |
| Personal monthly income, in Ringgit Malaysia | Mean (SD) | | 4094.9 (4128.6) | 4711.5 (5141.1) | 4246.2 (4403.3) |
| | Min to Max | | 500–60000 | 800–50000 | 500–60000 |
| | Median ± IQR | | 3000 ± 3000 | 3000 ± 3000 | 3000 ± 3000 |
| Alcohol, n (%) | Non-Alcohol drinker | | 597 (79.7) | 437 (81.8) | 1034 (80.6) |
| | Alcohol drinker | | 152 (20.3) | 97 (18.2) | 249 (19.4) |
| Smoke, n (%) | Non-smoker | | 630 (84.1) | 441 (82.6) | 1071 (83.5) |
| | Smoker | | 119 (15.9) | 93 (17.4) | 212 (16.5) |
| Vegetarian, n (%) | Not a vegetarian | | 736 (98.3) | 516 (96.6) | 1252 (97.6) |
| | Vegetarian | | 13 (1.7) | 18 (3.4) | 31 (2.4) |
| Hypertension, n (%) | | No | 699 (93.3) | 304 (56.9) | 1003 (78.2) |
| | | Yes | 50 (6.7) | 230 (43.1) | 280 (21.8) |
| Diabetes, n (%) | | No | 708 (94.5) | 409 (76.6) | 1117 (87.1) |
| | | Yes | 41 (5.5) | 125 (23.4) | 166 (12.9) |
| Neurological disorders, n (%)* | | No | 733 (97.9) | 509 (95.3) | 1242 (96.8) |
| | | Yes | 16 (2.1) | 25 (4.7) | 41 (3.2) |
| Had a family history of neurological disorders*, n (%) | | No | 693 (92.5) | 496 (92.9) | 1189 (92.7) |
| | | Yes | 56 (7.5) | 38 (7.1) | 94 (7.3) |
| Having diabetic related complications among those with diabetes, n (%) | Stroke | No | 40 (97.6) | 115 (92.0) | 155 (93.4) |
| | | Yes | 1 (2.4) | 10 (8.0) | 11 (6.6) |
| | Heart disease | No | 40 (97.6) | 100 (80.0) | 140 (84.3) |
| | | Yes | 1 (2.4) | 25 (20.0) | 26 (15.7) |
| | Renal disease | No | 39 (95.1) | 112 (89.6) | 151 (91.0) |
| | | Yes | 2 (4.9) | 13 (10.4) | 15 (9.0) |
| | Eye disease | No | 39 (95.1) | 84 (67.2) | 123 (74.1) |
| | | Yes | 2 (4.9) | 41 (32.8) | 43 (25.9) |
| | Foot ulceration | No | 38 (92.7) | 122 (97.6) | 160 (96.4) |
| | | Yes | 3 (7.3) | 3 (2.4) | 6 (3.6) |

*Guillain-Barre syndrome and chronic inflammatory demyelinating Polyneuropathy.

Malaysians having undiagnosed diabetes mellitus, likely due to the increasing obesity epidemic in Malaysia.

The study suggests that peripheral neuropathy can be caused by various medical conditions aside from diabetes mellitus, including age over 50, male gender, lower education, and

**Table 2. Percentage of peripheral neuropathy based on different types of tests.**

| Screening methods | Severity index | Aged ≤50 (n = 749) | Aged >50 (n = 534) | All |
|---|---|---|---|---|
| Biothesiometer | Normal, n (%) | 655 (87.4) | 288 (54.9) | 943 (73.5) |
| | Mild, n (%) | 60 (8.0) | 84 (15.7) | 114 (11.2) |
| | Moderate, n (%) | 15 (2.0) | 45 (8.4) | 60 (4.7) |
| | Severe, n (%) | 19 (2.5) | 117 (21.9) | 136 (10.6) |
| Neuropathy symptom scale | Normal, n (%) | 593 (79.2) | 324 (60.7) | 917 (71.5) |
| | Mild, n (%) | 71 (9.5) | 102 (19.1) | 173 (13.5) |
| | Moderate, n (%) | 70 (9.3) | 80 (15.0) | 150 (11.7) |
| | Severe, n (%) | 15 (2.0) | 28 (5.2) | 43 (3.4) |
| Neuropathy disability scale | Normal, n (%) | 719 (96.0) | 421 (78.8) | 1140 (88.9) |
| | Mild, n (%) | 25 (3.3) | 63 (11.8) | 88 (6.9) |
| | Moderate, n (%) | 5 (0.7) | 46 (8.6) | 51 (4.0) |
| | Severe, n (%) | 0 (0.0) | 4 (0.7) | 4 (0.3) |

The severity of peripheral neuropathy based on biothesiometer test was categorised as normal (1-15v), mild (16-20v), moderate (21-25v), and severe (26-50v) for those above 50 years old; for those 50 years old and below, normal is 1-10v, mild neuropathy (11-15v), moderate neuropathy (16-20v) and severe neuropathy (21-50v)

hypertension or diabetes. Subgroup analysis revealed different predictors for those with and without diabetes. For those without diabetes, the predictors were age over 50, male gender, other ethnicities, and low education. Meanwhile, among those with diabetes, the predictors were hypertension and a long history of diabetes.

The link between hypertension and peripheral neuropathy remains unclear. However, a study on rats found that those with hypertension developed nerve ischemia, thermal hyperalgesia, nerve conduction slowing, and axonal atrophy, in contrast to normotensive rats. The study suggested this could be due to demyelination and reduced levels of myelin essential protein in the nerves [28].

Peripheral neuropathy is known to be more prevalent in older adults, with a reported 8% of adults over 65 years experiencing some degree of neuropathy. This could be due to the higher ratio of macro and microvascular complications in older adults, as well as a higher prevalence of diabetes. In this study, the prevalence of diabetes was found to be 5.5% among those ≤50 years old and 23.4% among those >50 years old, with a higher percentage of males (16.4%) having diabetes compared to females (9.4%). Lower education was also identified as a predictor of peripheral neuropathy, possibly due to a higher percentage of those with lower education (secondary school and below) having diabetes (18.5%) compared to those with higher education (pre-university or tertiary education) at only 7.9%. Furthermore, a higher percentage of those with lower education were found to have hypertension (31.9%) compared to those with higher education at only 12.6%.

## Strength and limitations

This study has both strengths and limitations, as is common with most research studies. The strengths of the study are as follows,. To date, the study had a large sample size (n = 1283), which is beneficial in increasing the statistical power of the study and increasing the generalizability of the findings to the population. Second, the study was community-based, which means that the participants were representative of the general population, and the findings can be generalized to the general population. Third, the study used multiple logistic regression analysis, a statistical technique that allows for identifying predictors of a particular outcome while controlling for other variables. This method increases the accuracy and reliability of the findings.

**Table 3. Factors associated with severity of peripheral neuropathy using chi-square (n = 1283).**

| Variables | Category | Normal, 943 (73.5) | Mild-severe, 340 (26.5) | P-values |
|---|---|---|---|---|
| Age | ≤50 | 655 (87.4) | 94 (12.6) | <0.001 |
| | >50 | 288 (53.9) | 246 (46.1) | |
| Gender | Male | 448 (69.1) | 200 (30.9) | <0.001 |
| | Female | 495 (78.0) | 140 (22.0) | |
| Ethnicity | Malay | 302 (81.2) | 70 (18.8) | 0.001 |
| | Chinese | 492 (70.9) | 202 (29.1) | |
| | Indian | 111 (68.9) | 50 (31.1) | |
| | Others | 38 (67.9) | 18 (32.1) | |
| Education | None | 16 (41.0) | 240 (39.3) | <0.001 |
| | Primary | 54 (42.9) | | |
| | Secondary | 301 (67.5) | | |
| | Pre-U | 96 (83.5) | 100 (14.9) | |
| | Tertiary | 476 (85.5) | | |
| Drink Alcohol | No | 757 (73.2) | 277 (26.8) | 0.633 |
| | Yes | 186 (74.7) | 63 (25.3) | |
| Smoking | No | 802 (74.9) | 269 (25.1) | 0.012 |
| | Yes | 141 (66.5) | 71 (33.5) | |
| Vegetarian | No | 923 (73.7) | 329 (26.3) | 0.251 |
| | Yes | 20 (64.5) | 11 (35.5) | |
| Having hypertension | No | 814 (81.2) | 189 (18.8) | <0.01 |
| | Yes | 129 (46.1) | 151 (53.9) | |
| Having diabetes | No | 888 (79.5) | 229 (20.5) | <0.01 |
| | Yes | 55 (33.1) | 111 (66.9) | |
| Having other neurological disease | No | 918 (73.9) | 324 (26.1) | 0.065 |
| | Yes | 25 (61.0) | 16 (39.0) | |
| Having family history of neurological disease | No | 873 (73.4) | 316 (26.6) | 0.825 |
| | Yes | 70 (74.5) | 24 (25.5) | |

The peripheral neuropathy was classified as "yes" for individuals with mild to severe peripheral neuropathy, while "no" referred to those with normal conditions.

This study must be interpreted with a few limitations. First, the study had a cross-sectional design, and the researchers could not examine the causality between the predictors and the outcome. Second, participants were recruited from seven retail pharmacies located in the state of Selangor, which may limit the generalizability of the findings to other populations or settings. Third, the study relied on self-reported data, which may be subject to bias, as participants may not have accurately reported their medical history or symptoms. Fourth, the study did not capture all potentially relevant variables, such as BMI, pre-diabetes status, and prior diagnosis of peripheral neuropathy, which may limit the accuracy and precision of the findings.

In summary, this cross-sectional study has some notable strengths, such as the large sample size and the use of multiple logistic regression analysis. However, it has several limitations, including the cross-sectional design, selection bias, information bias, and limited variables.

The Malay ethnic group constitutes the majority in Malaysia, but in this study, Chinese respondents accounted for 54.1%. The predominance of Chinese ethnic respondents in our study can be attributed to several factors. Chinese Malaysians constitute a significant segment of the urban population, where retail pharmacies are mainly concentrated. Furthermore, cultural attitudes and socioeconomic status may influence the likelihood of individuals within the

**Table 4. Predictors of peripheral neuropathy using multiple logistic regression (n = 1283).**

| Characteristics | Categories | Overall (n = 1283) | | | |
|---|---|---|---|---|---|
| | | Adjusted OR | Lower CI | Upper CI | P-value |
| Having diabetes | No | Reference | Reference | Reference | Reference |
| | Yes | 3.901 | 2.610 | 5.830 | <0.001 |
| Age | ≤50 | Reference | Reference | Reference | Reference |
| | >50 | 3.376 | 2.447 | 4.658 | <0.001 |
| Education | Pre-U/Tertiary | Reference | Reference | Reference | Reference |
| | None/Primary/Secondary | 2.330 | 1.709 | 3.178 | <0.001 |
| Gender | Female | Reference | Reference | Reference | Reference |
| | Male | 1.816 | 1.354 | 2.436 | <0.001 |
| Ethnicity | Malay | Reference | Reference | Reference | Reference |
| | Chinese | 0.976 | 0.670 | 1.422 | 0.899 |
| | Indian | 1.216 | 0.738 | 2.005 | 0.442 |
| | Others | 1.867 | 0.931 | 3.746 | 0.079 |
| Having hypertension | No | Reference | Reference | Reference | Reference |
| | Yes | 1.662 | 1.174 | 2.351 | 0.004 |
| Having other neurological disease | No | Reference | Reference | Reference | Reference |
| | Yes | 1.289 | 0.588 | 2.828 | 0.526 |
| Smoke | No | Reference | Reference | Reference | Reference |
| | Yes | 1.029 | 0.699 | 1.516 | 0.883 |

Multiple logistic regression analysis for overall samples, the omnibus tests of model coefficients for logistic regression analysis were statistically significant (p<0.05) with the chi-square in the final model, $X^2$ = 312.901 (7), p <0.001. This indicates that the model is good at predicting the outcome than it was with the baseline model when only the constant is included. Meanwhile, the Nagelkerke ($R^2$) was 0.316, which means that 31.6% of the variation in the outcome is explained by the independent variables. Hosmer-Lemeshow goodness of fit also showed that the data is fit for the model with p>0.05, $X^2$ = 11.978. The model was 77.7% accurate in its prediction of peripheral neuropathy among the participants.

Chinese community to seek health services from retail pharmacies. This tendency could be related to greater health awareness, accessibility, and a preference for retail pharmacies for minor health concerns among this ethnic group. It is important to note that the distribution of ethnic groups in this study does not accurately reflect the actual demographic proportions in Malaysia; rather, it is a consequence of employing a non-probability sampling method. These limitations should be considered when interpreting the findings of this study, and future studies should aim to address these limitations.

Identifying diabetes as a significant predictor of peripheral neuropathy highlights the importance of understanding the mechanisms underlying the relationship between these two conditions. It is well known that diabetes can cause nerve damage, but the specific pathways involved are not fully understood. Further research in this area may lead to the development of new treatments that target these mechanisms. Similarly, the identification of age, education, gender, and hypertension as predictors of peripheral neuropathy suggest that there may be complex interactions between these factors and the nervous system. For example, age-related changes in the nervous system may make older individuals more susceptible to peripheral neuropathy. At the same time, education and gender may influence lifestyle factors such as diet and exercise that can affect nerve health [29].

To our knowledge, this is the first comprehensive study among the prevalence and the factors of peripheral neuropathy among retail pharmacies in Malaysia using a biothesiometer. Overall, the findings presented in the results highlighted the importance of studying peripheral neuropathy from a multidisciplinary perspective, incorporating insights from neuroscience,

endocrinology, and other fields. By identifying the underlying mechanisms and risk factors associated with this condition, this study signifies that there is a need for targeted public health campaigns aimed at increasing awareness about the condition, its risk factors, and early symptoms, which encompass diverse ethnic groups and emphasize the importance of regular health check-ups, particularly for individuals with comorbidities such as diabetes and hypertension. Furthermore, healthcare providers play a crucial role in public awareness, and training programs for healthcare professionals should thus emphasize the significance of early detection, prompt intervention, and patient education regarding peripheral neuropathy. This, in turn, may contribute to improved patient outcomes and a reduced burden on healthcare systems.

## Conclusion

Around one-quarter of the general population demonstrated mild to severe symptoms of peripheral neuropathy screened by biothesiometer. The percentage was higher among those >50 years old (45.1%) than those ≤50 years old (12.6%). Those with hypertension and diabetes, aged >50 years old and male, and with secondary education or below are associated with mild to severe symptoms of peripheral neuropathy. Clinicians should be made aware of the high prevalent of PN in general populations and should make regular screening, especially among patients with risk factors (i.e. diabetes and hypertension) to prevent associated complications.

## Supporting information

**S1 Table. Neuropathy symptom score.**
(DOCX)

**S2 Table. Neuropathy disability score.**
(DOCX)

## Acknowledgments

We thank the retailed pharmacies which allowed this study to be conducted within their premises and proactively supported us throughout the screening and awareness campaign. We also want to thank medical graduates (Dr. Nurul Mursyidah Zakaria, Dr. Reynart Chow Wei Yong, and Dr. Yee Ling Shin) and Mr Poon Khai Lang for your effort in recruiting participants for this study.

## Author Contributions

**Conceptualization:** Siew Mooi Ching, Kai Wei Lee, Abdul Hanif Khan Yusof Khan, Navin Kumar Devaraj, Ai Theng Cheong, Fan Kee Hoo, Wan Aliaa Wan Sulaiman, Wei Chao Loh.

**Data curation:** Navin Kumar Devaraj, Ai Theng Cheong, Fan Kee Hoo, Wan Aliaa Wan Sulaiman, Wei Chao Loh.

**Funding acquisition:** Siew Mooi Ching.

**Methodology:** Siew Mooi Ching, Kai Wei Lee, Abdul Hanif Khan Yusof Khan, Navin Kumar Devaraj.

**Project administration:** Kai Wei Lee, Shen Horng Chong.

**Resources:** Siew Mooi Ching, Fan Kee Hoo, Mansi Patil.

**Software:** Kai Wei Lee.

**Supervision:** Siew Mooi Ching, Ai Theng Cheong, Fan Kee Hoo.

**Validation:** Wan Aliaa Wan Sulaiman, Wei Chao Loh.

**Visualization:** Siew Mooi Ching, Kai Wei Lee.

**Writing – original draft:** Siew Mooi Ching, Kai Wei Lee, Sook Fan Yap.

**Writing – review & editing:** Siew Mooi Ching, Kai Wei Lee, Abdul Hanif Khan Yusof Khan, Navin Kumar Devaraj, Ai Theng Cheong, Sook Fan Yap, Fan Kee Hoo, Wan Aliaa Wan Sulaiman, Wei Chao Loh, Mansi Patil, Vasudevan Ramachandran.

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
