## [Decision Letter · Decision Letter 0]

11 Dec 2023

PONE-D-23-27646Prevalence and factor associated with peripheral neuropathy in a setting of retail pharmacies in Malaysia – A cross-sectional studyPLOS ONE

Dear Dr. Ching,

Thank you for submitting your manuscript to PLOS ONE. After careful consideration, we feel that it has merit but does not fully meet PLOS ONE’s publication criteria as it currently stands. Therefore, we invite you to submit a revised version of the manuscript that addresses the points raised during the review process.

We look forward to receiving your revised manuscript.

Kind regards,

Ismail Tawfeek Abdelaziz Badr, M.D.

Academic Editor

PLOS ONE

“Prof. Dr. Siew Mooi Ching

Funded by Procter & Gamble (M) Sdn. Bhd (Vote ID: 6380068).”

Additional Editor Comments:

The great discrepancy of the prevalence of peripheral neuropathy compared to what reported to the literature needs more explanation. why the incidence might be high in your selected population ?

update of reference to a more recent group is required

Reviewers' comments:

Reviewer's Responses to Questions

**Comments to the Author**

1. Is the manuscript technically sound, and do the data support the conclusions?

Reviewer #1: Yes

Reviewer #2: No

2. Has the statistical analysis been performed appropriately and rigorously? 

Reviewer #1: Yes

Reviewer #2: Yes

3. Have the authors made all data underlying the findings in their manuscript fully available?

Reviewer #1: Yes

Reviewer #2: Yes

4. Is the manuscript presented in an intelligible fashion and written in standard English?

Reviewer #1: Yes

Reviewer #2: No

5. Review Comments to the Author

Reviewer #1: Overall, well done and a good read. Here are a few comments to consider.

Line 100

Comparison between previous studies, in reference 12-13 are from years 2003-2012. Is there any newer study?

Reference 15 is on painful neuropathy from 2017, showing a lesser prevalence than references 12-13; is there any newer study? If not, perhaps can mention, “in the absence of other studies, the latest prevalence was... in year ... “

Line 108

Is there any reference for poor awareness among patients and less accessibility to gain proper attention from physicians? How about PHC? In Malaysia, PHC teams have been trained to identify DM wounds and have dedicated DM clinics and wound care teams. So, should be careful with these kinds of strong statements. It seems to pinpoint the reason for DM wounds is due to lack of access to treatment.

Line 125 Methods

How come the sample is taken as those aged more than 18? Most diabetics are older.

Where was this convenient sampling done? Only later is it mentioned as a single location. Why was it not done in more locations? Is there data to support that by taking a sample from one location, it can be generalised to all over Malaysia? Can at least mention which location it was.

Line 132 sample size calculation

Any justification on similarity of Parsis in Mumbai/Bombay and Malaysia?

Line 150 biothesiometer usage, line 166 examination

What if they had a wound/ ulcer on the foot? Were they excluded?

Line 183 Statistical analysis

How about other comorbids eg hypercholesterolemia /IHD?

The outcome variable, severity of peripheral neuropathy is only explained in next section (line 194, operational definition), whereas here under statistical analysis, scoring here and in methods was for neuro symptoms and neuro disability. Perhaps could be made a little clearer.

Line 212 baseline characteristics

If we are asking about diet (nonveg), how about hyperlipidemia? Especially if can ask about DM and HPT (self-reported).

Line 305 Limitations

single location- have mentioned as above

Line 331

Retail pharmacists or pharmacies?

Line 337

Good public health implications and inferences.

Line 475 tables

Perhaps can rewrite min – max as min to max, so as not to denote a mathematical equation.

Ethnicity, how come Chinese are highest? Whereas Malays are the largest ethnic group? Is there any reason that can explain this finding. If so, should elaborate in the discussion.

Reviewer #2: The manuscript needs some technical modifications.

The statistical analysis has been performed appropriately but the version mentioned in the abstract is not the same as statistical analysis.

The data are available upon request.

The manuscript needs good language , English editing, and Grammer correction.

The explanation of prepheral neuropathy in hypertension is not clear and has no supportive studies on human beings.

The prevelance of prepheral neuropathy has a very high percentage , more than the quarter, please mention clear explanation.

The exclusion criteria is not satisfactory and doesn't exclude all conditions that may cause prepheral neuropathy.

6. PLOS authors have the option to publish the peer review history of their article (what does this mean?). If published, this will include your full peer review and any attached files.

Reviewer #1: No

Reviewer #2: No

---

## [Author Response · Author response to Decision Letter 0]

27 Apr 2024

Thank you for the comments to improvise the manuscript, we have revised the manuscript based on the comments provided and used the track changes for revision. 

Reviewer Comments

1. Line 100 - Comparison between previous studies, in reference 12-13 are from years 2003-2012. Is there any newer study? Reference 15 is on painful neuropathy from 2017, showing a lesser prevalence than references 12-13; is there any newer study? If not, perhaps can mention, “in the absence of other studies, the latest prevalence was... in year ... “

Author response: We found two meta-analysis report prevalence of DPN among T1DM, T2DM and pre-diabetes. We have inserted the two reviews in the paragraph, to read it as: 

“Overall, the finding from a recent meta-analysis showed that the pooled prevalence of DPN was 30% and DPN is more prevalent in people with T2DM (31.5%) compared to those in T1DM (17.5%) [15]. Prevalence of peripheral neuropathy in patients with diabetes: A systematic review and meta-analysis. Primary care diabetes, 14(5), 435-444). Another meta-analysis also report that the DPN was also common (>10%) among those with pre-diabetes [16].

Reviewer Comments

2. Line 108 - Is there any reference for poor awareness among patients and less accessibility to gain proper attention from physicians? How about PHC? In Malaysia, PHC teams have been trained to identify DM wounds and have dedicated DM clinics and wound care teams. So, should be careful with these kinds of strong statements. It seems to pinpoint the reason for DM wounds is due to lack of access to treatment. 

Author response: Sorry for the confusion, we have replaced that sentence with another one, to read it as: 

“The diagnosis of peripheral neuropathy has been overlooked especially among the general population, due to poor awareness among patients and less accessibility to gain proper attention from physicians for early symptoms presentation. Lack of widespread awareness about peripheral neuropathy, coupled with its subtle symptoms that don't significantly disrupt daily life, often results in individuals with this condition neglecting to seek medical attention for these minor manifestations.”

Reviewer Comments

3. Line 125 Methods - How come the sample is taken as those aged more than 18? Most diabetics are older. Where was this convenient sampling done? Only later is it mentioned as a single location. Why was it not done in more locations? Is there data to support that by taking a sample from one location, it can be generalised to all over Malaysia? Can at least mention which location it was. 

Author response – This was a community study involving 7 retail pharmacies. (refer Study design, study population and sampling method). The objective of this study was to establish the prevalence of peripheral neuropathy among the general population. Therefore, we included those aged ≥18 years old with convenient sampling method, regardless of diabetes status. The community outreach's purpose is to increase awareness, not just focusing on peripheral neuropathy among people with diabetes. We have made necessary amendment in limitation section, to read it as: 

“This study must be interpreted with a few limitations. First, the study had a cross-sectional design, which means that it was conducted at a single point in time, and the researchers could not examine the causality between the predictors and the outcome. Second, the participants were recruited from a single location retailed pharmacies, which may limit the generalizability of the findings to other populations or settings. Third, the study relied on self-reported data, which may be subject to bias, as participants may not have accurately reported their medical history or symptoms. Fourth, the study did not capture all potentially relevant variables, such as BMI, pre-diabetes status, and prior diagnosis of peripheral neuropathy, which may limit the accuracy and precision of the findings.”

Reviewer Comments

4. Line 132 sample size calculation - Any justification on similarity of Parsis in Mumbai/Bombay and Malaysia? 

Author response: I agree with you that there could be no similarity between Parsis (Mumbai) and Malaysian population, but that study (20.Bharucha NE, Bharucha AE, Bharucha EP. Prevalence of peripheral neuropathy in the Parsi community of Bombay. Neurology. 1991 Aug 1;41(8):1315-.) is the only community study involved general public. Therefore we used it as the reference for sample size calculation. In addition, For this study, a non-probability sampling method was employed. The application of traditional sample size calculations in such scenarios aimed to ensure a sufficient amount of data, yielding reasonable precision and meaningful outputs within the study's constraints. Sample size calculation was more crucial to a study conducted with a probability sampling method.

Reviewer Comments

5. Line 150 - biothesiometer usage, line 166 examination. What if they had a wound/ ulcer on the foot? Were they excluded? 

Author response – Exclusion was not applied to individuals with foot ulceration or wounds. The data pertaining to "Diabetic-related complications among individuals with diabetes" had been shown in Table 1. 

Reviewer Comments

6. Line 183 Statistical analysis - How about other co-morbids eg hypercholesterolemia /IHD? 

Author response – 

In response to the inquiry about including other comorbidities like hypercholesterolemia and ischemic heart disease (IHD), we value the reviewer's suggestion. However, it's important to note that our data collection sheet did not encompass these specific comorbidities. Our primary focus was on variables directly pertinent to the study's objective, which is to investigate the prevalence and factors associated with peripheral neuropathy in the community level. 

Even though we had collected the diabetic-related complications like IHD but the sample size was small (n=1) as shown in Table 1. Nevertheless, we acknowledge the merit of a more comprehensive approach in future research endeavors. We appreciate the suggestion and plan to consider the incorporation of additional comorbidities, including hypercholesterolemia and IHD, to enrich the depth of our analysis. This approach will contribute to a more thorough examination of potential associations in our future studies.

Reviewer Comments

7. The outcome variable, severity of peripheral neuropathy is only explained in next section (line 194, operational definition), whereas here under statistical analysis, scoring here and in methods was for neuro symptoms and neuro disability. Perhaps could be made a little clearer. 

Author response – Sorry for the confusion, we have amended the sentence, to read it as: 

The dependent variable, categorized as "yes" in the presence of mild to severe symptoms of peripheral neuropathy, underwent testing through either the Chi-Square test or Fisher’s exact test. Variables with a p-value < 0.25 in the simple logistic regression (derived from the results in the normal versus mild-severe column) were entered into multiple logistic regressions to look for determinants of peripheral neuropathy (Mild-severe). The level of significance is set at a p-value < 0.05.

Reviewer Comments

8. Line 212 baseline characteristics. If we are asking about diet (nonveg), how about hyperlipidemia? Especially if can ask about DM and HPT (self-reported). 

Author response - We acknowledge the reviewer's suggestion to include hyperlipidemia in the baseline characteristics, particularly in relation to inquiries about a nonvegetarian diet. While we collected self-reported information on diabetes (DM) and hypertension (HPT) due to their association with peripheral neuropathy, hyperlipidemia was not included in the data collection sheet as the correlation between statin therapy and peripheral neuropathy is a matter of debate.(reference: Pasha R, Azmi S, Ferdousi M, Kalteniece A, Bashir B, Gouni-Berthold I, Malik RA, Soran H. Lipids, Lipid-Lowering Therapy, and Neuropathy: A Narrative Review. Clin Ther. 2022 Jul;44(7):1012-1025. doi: 10.1016/j.clinthera.2022.03.013. Epub 2022 Jul 6. PMID: 35810030. 

However, we acknowledge the significance of assessing hyperlipidemia in relation to dietary habits and will consider including this variable in future research to provide a more comprehensive understanding of baseline characteristics.

Reviewer Comments

9. Line 305 Limitations - single location- have mentioned as above 

Author response – We have amended the typo, to read it as: 

“This study must be interpreted with a few limitations. First, the study had a cross-sectional design, which means that it was conducted at a single point in time, and the researchers could not examine the causality between the predictors and the outcome. Second, the participants were recruited from a single location retailed pharmacies, which may limit the generalizability of the findings to other populations or settings. Third, the study relied on self-reported data, which may be subject to bias, as participants may not have accurately reported their medical history or symptoms. Fourth, the study did not capture all potentially relevant variables, such as BMI, pre-diabetes status, and prior diagnosis of peripheral neuropathy, which may limit the accuracy and precision of the findings.”

Reviewer Comments

10. Line 331 - Retail pharmacists or pharmacies?

Author response – We have amended the typo, to read it as: 

“To our knowledge, this is the first comprehensive study among the prevalence and the factors of peripheral neuropathy among retail pharmacists pharmacies in Malaysia using a biothesiometer.”

Reviewer Comments

11. Line 337 - Good public health implications and inferences. 

Author response – Thank you for your enlightenment, we have strengthened the implications with the following changes, to read it as: 

“By identifying the underlying mechanisms and risk factors associated with this condition, researchers can develop more effective treatments and preventative measures to improve the quality of life for those affected by peripheral neuropathy this study signifies that there is a need for targeted public health campaigns aimed at increasing awareness about the condition, its risk factors, and early symptoms, which encompass diverse ethnic groups and emphasize the importance of regular health check-ups, particularly for individuals with comorbidities such as diabetes and hypertension. Furthermore, healthcare providers play a crucial role in public awareness, such as training programs for healthcare professionals could emphasize the significance of early detection, prompt intervention, and patient education regarding peripheral neuropathy. This, in turn, can contribute to improved patient outcomes and a reduced burden on healthcare systems.”

Reviewer Comments

12. Line 475 tables - Perhaps can rewrite min – max as min to max, so as not to denote a mathematical equation. 

Respond: Amendment done.

Reviewer Comments

13. Line 475 Table:Ethnicity, how come Chinese are highest? Whereas Malays are the largest ethnic group? Is there any reason that can explain this finding. If so, should elaborate in the discussion. 

Author response – We have addressed this finding under limitation (line 332) to read as:

The Malay ethnic group constitutes the majority in Malaysia, but in this study, Chinese respondents accounted for 54.1%. This discrepancy may be attributed to a higher prevalence of Chinese clients at the selected retail pharmacies and their increased interest in assessing their peripheral neuropathy status. It's important to note that the distribution of ethnic groups in this study does not accurately reflect the actual demographic proportions in Malaysia; rather, it is a consequence of employing a non-probability sampling method.

---

## [Decision Letter · Decision Letter 1]

23 May 2024

PONE-D-23-27646R1Prevalence and factor associated with peripheral neuropathy in a setting of retail pharmacies in Malaysia – A cross-sectional studyPLOS ONE

Dear Dr. Ching,

Thank you for submitting your manuscript to PLOS ONE. After careful consideration, we feel that it has merit but does not fully meet PLOS ONE’s publication criteria as it currently stands. Therefore, we invite you to submit a revised version of the manuscript that addresses the points raised during the review process.

We look forward to receiving your revised manuscript.

Kind regards,

Ismail Tawfeek Abdelaziz Badr, M.D.

Academic Editor

PLOS ONE

Journal Requirements:

Reviewers' comments:

Reviewer's Responses to Questions

**Comments to the Author**

1. If the authors have adequately addressed your comments raised in a previous round of review and you feel that this manuscript is now acceptable for publication, you may indicate that here to bypass the “Comments to the Author” section, enter your conflict of interest statement in the “Confidential to Editor” section, and submit your "Accept" recommendation.

Reviewer #1: (No Response)

Reviewer #2: All comments have been addressed

2. Is the manuscript technically sound, and do the data support the conclusions?

Reviewer #1: Yes

Reviewer #2: No

3. Has the statistical analysis been performed appropriately and rigorously? 

Reviewer #1: Yes

Reviewer #2: No

4. Have the authors made all data underlying the findings in their manuscript fully available?

Reviewer #1: Yes

Reviewer #2: Yes

5. Is the manuscript presented in an intelligible fashion and written in standard English?

Reviewer #1: No

Reviewer #2: No

6. Review Comments to the Author

**Reviewer #1:** The authors have mostly addressed the queries put forth earlier. Just a few to improve the manuscript before publication.

Minor comments on:

Intro Line 98: Universiti Hospital – should this be spelt University Hospital, or University Malaya Medical Centre? If it is not UMMC, then should clearly identify which hospital this is, without reader having to view the reference.

Intro line 101: USM – pls write in full

Intro overall seems to focus on DM neuropathy. What about other types of neuropathy?

Sample size calculation lines 133-137: should add what you had explained in the rebuttal to my earlier comment in the first review about why you are comparing/ referring to a study conducted among the Parsi community in Mumbai, India, so that the reader understands your rationale as well.

Discussion line 273 grammatical correction, should read as, “The higher prevalence of peripheral

neuropathy in our study population compared to global studies could be attributed to the

biothesiometer's higher sensitivity in detecting early or mild peripheral neuropathy, as well as

the possibility of many Malaysians having undiagnosed diabetes mellitus, on par with the increasing

obesity epidemic in Malaysia.”

Discussion line 278- since you mention other ethnicities as significant for those aged >50, I feel you should substantiate the majority of Chinese ethnic respondents, as mentioned in your rebuttal to my my first review of your draft. It is clearly mentioned there, so do incorporate the baseline discussion of your findings in your paragraph, or at least why the majority are Chinese here.

Strengths and limitations line 309: when you mention single location, do elaborate- do you mean single state/ single region? Sounds better than one location alone which sounds like the researchers did not put in more effort, whereas you have actually gone to 7 different pharmacies.

Line 332 – I think you mean retail pharmacies, otherwise it would mean you are studying the retail pharmacists as your sample

Line 333 – the findings presented in the results (add the s)

Conclusion line 345 – to prevent associated complications.

**Reviewer #2: **I recommend adjusting the title and add the word factors instead of factor.

The sample size calculation isn't accurate.

The tables need to be more justified.

English editing is needed.

7. PLOS authors have the option to publish the peer review history of their article (what does this mean?). If published, this will include your full peer review and any attached files.

Reviewer #1: No

Reviewer #2: No

---

## [Author Response · Author response to Decision Letter 1]

27 Jun 2024

Journal Requirements:

Answer: We have made the changes accordingly to all the references

Review Comments to the Author

Reviewer #1: The authors have mostly addressed the queries put forth earlier. Just a few to improve the manuscript before publication.

Minor comments on:

Comment 1 

Intro Line 98: Universiti Hospital – should this be spelt University Hospital, or University Malaya Medical Centre? If it is not UMMC, then should clearly identify which hospital this is, without reader having to view the reference.

Answer: We have made the changes to read as” University Kebangsaan Malaysia Medical Center” (page 4, line 96)

Comment 2 

Intro line 101: USM – pls write in full

Answer: We have revised the sentences (page 5, line 101)

Comment 3

Intro overall seems to focus on DM neuropathy. What about other types of neuropathy?

Answer: Thank you for your input. We acknowledge that there are other types of neuropathy, such as Guillain-Barré syndrome (GBS), chronic inflammatory demyelinating polyneuropathy (CIDP), amyloid neuropathy, and acute inflammatory neuropathies. However, given the high prevalence of diabetes in Malaysia, diabetic neuropathy is likely the most common form of neuropathy in the general population, making it a significant early complication of diabetes. Therefore, our focus remains on diabetic neuropathy due to its prevalence and the potential for early detection.

We have added the following paragraph in the main text, to read it as: (page 5, line 115 to 126)

Diabetes mellitus is a highly prevalent condition in the Malaysian population, and diabetic neuropathy is one of the earliest and most common complications arising from it. As a large segment of the population is affected by diabetes, it is reasonable to assume that diabetic neuropathy is likely the most common form of neuropathy encountered in the general population. In contrast, other types of neuropathies mentioned, such as Guillain-Barré syndrome (GBS), chronic inflammatory demyelinating polyneuropathy (CIDP), amyloid neuropathy, and acute inflammatory neuropathies, are relatively rare and often present with rapidly progressive or severe symptoms that would prompt individuals to seek medical attention promptly. These forms of neuropathy are typically more severe and require immediate medical intervention, making them less relevant for the specific focus of this paper, which primarily addresses early detection and monitoring of neuropathy in the general population [15,16,17].

Comment 4: Sample size calculation lines 133-137: should add what you had explained in the rebuttal to my earlier comment in the first review about why you are comparing/ referring to a study conducted among the Parsi community in Mumbai, India, so that the reader understands your rationale as well.

Answer: We have added this sentence to read as (page 7, line 165-171)

While determining the appropriate sample size for our study on the prevalence of diabetic neuropathy in the community, we encountered a scarcity of recent, locally relevant research to serve as a reference. The paucity of community-based neuropathy studies, particularly in our region, necessitated the use of an older study conducted in Mumbai, India, in 1991 [22]. Despite its age, this study remains one of the few comprehensive investigations into the prevalence of neuropathy in a community setting, making it a valuable resource for our sample size calculations.

Comment 5: This integration maintains the focus of your study while explaining the rationale behind your methodological choices.

We have added this in the main text to read as: (page 6-7; line 144-155)

Retail pharmacies were selected as data collection sites based on several considerations. Firstly, individuals with chronic diseases, such as diabetes, frequently visit retail pharmacies to obtain their medications or seek alternative therapeutic options like dietary supplements. These individuals are already engaged with pharmacists in managing their health conditions. By conducting this study at retail pharmacies, we could access a significant portion of the target population who are likely to be more health-literate and inclined to learn about their health status.

Additionally, we conducted this study in the form of a health campaign offering free neuropathy screenings, which further facilitated convenient data collection. This approach not only benefits those with diabetes by providing them with an opportunity to assess their neuropathy status, but also allows individuals without diabetes to undergo screening and increase their awareness about neuropathy. Furthermore, the health campaign raised awareness among the public visiting the pharmacy, including family members or companions of those with diabetes.

Comment 6: Discussion line 273 grammatical correction, should read as, “The higher prevalence of peripheral neuropathy in our study population compared to global studies could be attributed to the biothesiometer's higher sensitivity in detecting early or mild peripheral neuropathy, as well as

the possibility of many Malaysians having undiagnosed diabetes mellitus, on par with the increasing obesity epidemic in Malaysia.”

Answer: We have amended the paragraph as suggested page 13, line 303 to 307.

Comment 7: Discussion line 278- since you mention other ethnicities as significant for those aged >50, I feel you should substantiate the majority of Chinese ethnic respondents, as mentioned in your rebuttal to myfirst review of your draft. It is clearly mentioned there, so do incorporate the baseline discussion of your findings in your paragraph, or at least why the majority are Chinese here.

Answer: Thank you for your observation. The majority of Chinese ethnic respondents in retail pharmacies can be attributed to several factors. In Malaysia, Chinese Malaysians represent a significant proportion of the urban population, where retail pharmacies are predominantly located. Additionally, cultural and socioeconomic factors may contribute to a higher propensity among the Chinese community to seek health services from retail pharmacies.

We have added the following explanation to the discussion: (page15, line 351-362)

The Malay ethnic group constitutes the majority in Malaysia, but in this study, Chinese respondents accounted for 54.1%. The predominance of Chinese ethnic respondents in our study can be attributed to several factors. Chinese Malaysians constitute a significant segment of the urban population, where retail pharmacies are mainly concentrated. Furthermore, cultural attitudes and socioeconomic status may influence the likelihood of individuals within the Chinese community to seek health services from retail pharmacies. This tendency could be related to greater health awareness, accessibility, and a preference for retail pharmacies for minor health concerns among this ethnic group. It is important to note that the distribution of ethnic groups in this study does not accurately reflect the actual demographic proportions in Malaysia; rather, it is a consequence of employing a non-probability sampling method. These limitations should be considered when interpreting the findings of this study, and future studies should aim to address these limitations.

Comment 8: Strengths and limitations line 309: when you mention single location, do elaborate- do you mean single state/ single region? Sounds better than one location alone which sounds like the researchers did not put in more effort, whereas you have actually gone to 7 different pharmacies.

Answer: That was referring to single state with seven retail pharmacies. We have revised the sentence to read as (page 14, line 340-343 ):

This study must be interpreted with a few limitations. First, the study had a cross-sectional design, and the researchers could not examine the causality between the predictors and the outcome. Second, participants were recruited from seven retail pharmacies located in the state of Selangor, which may limit the generalizability of the findings to other populations or settings.

Comment 9: Line 332 – I think you mean retail pharmacies, otherwise it would mean you are studying the retail pharmacists as your sample. 

Answer: yes. You are right. Correction had been made.

Comment 10: Line 376 – the findings presented in the results (add the s)

Answer: We have made the amendment as suggested (Line 376)

Comment 11: Conclusion line 394 – to prevent associated complications.

Answer: We have made the amendment as suggested (Line 394)

Reviewer #2: 

Comment 1: I recommend adjusting the title and add the word factors instead of factor.

Answer: We have amended the title accordingly to read as

“Prevalence and factors associated with peripheral neuropathy in a setting of retail pharmacies in Malaysia – A cross-sectional study”

Comment 2: The sample size calculation isn't accurate.

Answer: We acknowledge that the optimal approach for calculating the sample size would have been based on the two-proportion formula, considering the current study aims to identify factors associated with peripheral neuropathy. The initial sample size was calculated using the StatCalc function in Epi Info 7.0, based on the prevalence of peripheral neuropathy among the Parsi community of Bombay, which was 2.38% [22]. This calculation resulted in an estimated sample size of 899, with a 95% confidence interval (CI), 80% power, and a p-value of <0.05. After accounting for a potential missing value rate of 30%, the total required number of participants was determined to be 1,283.

We did not use the prevalence of peripheral neuropathy from studies conducted at UKM, UM, or USM, as these studies were among diabetic patients in outpatient clinics. This setting could lead to selection bias, as patients at outpatient clinics are more likely to have advanced or severe forms of the condition, resulting in an overestimation of prevalence. Such overestimation could inflate the sample size requirement, not accurately reflecting the true population parameters. Therefore, we adhered to the original reference for sample size calculation, which was the Mumbai community study. My study aims to assess the prevalence and associated factors of peripheral neuropathy in a community setting, similar to the Mumbai study. Using a reference study with a comparable population and setting ensures greater alignment with the research objectives and enhances the validity of the sample size calculation.

For the factors associated with peripheral neuropathy, we used G*Power software, which calculated a required sample size of 8,422. This calculation was based on a study conducted at Universiti Kebangsaan Malaysia, where older age was identified as a determinant of diabetic peripheral neuropathy among patients with diabetes (OR 1.13, 95% CI 1.01-1.26, p=0.039) [22]. Due to the impractically large sample size and time constraints, we adopted a sample size of 899 for this study.

Reference: Faul, F., Erdfelder, E., Lang, A.-G., & Buchner, A. (2007). G*Power 3: A flexible statistical power analysis program for the social, behavioral, and biomedical sciences. Behavior Research Methods, 39, 175-191.).

We have added statement under sample size calculation session (line 172-177)

For the factors associated with peripheral neuropathy, we used G*Power software, which calculated a required sample size of 8,422. This calculation was based on a study conducted at Universiti Kebangsaan Malaysia, in which older age was identified as a determinant of diabetic peripheral neuropathy among patients with diabetes (OR 1.13, 95% CI 1.01-1.26, p=0.039) [13]. Due to the impractically large sample size and time constraints, we adopted a sample size of 899 for this study.

Comment 3: The tables need to be more justified.

Answer: We have simplified the tables now.

Table 1: Characteristics of participants’ socio-demographics, lifestyle, medical background, family history, and diabetic-related complications (n=1283)-remained

Table 2: Percentage of peripheral neuropathy based on different tests still remained

Table 3 on Percentage of peripheral neuropathy (screened by biothesiometer) according to age groups have been deleted from the main text

Table 4 becomes Table 3 and the presentation has been simplified 

Table 5 becomes Table 4 for predictors of peripheral neuropathy using multiple logistic regression (n=1283)

Comment 4: English editing is needed.

Answer: We have sent the manuscript for English editing as suggested and attached the certificate of proof reading.

---

## [Editor Report · Decision Letter 2]

1 Jul 2024

Prevalence and factor associated with peripheral neuropathy in a setting of retail pharmacies in Malaysia: A cross-sectional study

PONE-D-23-27646R2

Dear Dr. Ching,

We’re pleased to inform you that your manuscript has been judged scientifically suitable for publication and will be formally accepted for publication once it meets all outstanding technical requirements.

Kind regards,

Academic Editor

PLOS ONE

---

## [Editor Report · Acceptance letter]

16 Oct 2024

PONE-D-23-27646R2 

PLOS ONE

Dear Dr. Ching, 

I'm pleased to inform you that your manuscript has been deemed suitable for publication in PLOS ONE. Congratulations! Your manuscript is now being handed over to our production team.

Kind regards, 

on behalf of

Dr. Ismail Tawfeek Abdelaziz Badr 

Academic Editor

PLOS ONE